# Cohort profile: the Western Cape Pregnancy Exposure Registry (WCPER)

Emma Kalk ,[1] Alexa Heekes,[1,2] Amy L Slogrove,[3,4] Florence Phelanyane,[1,2] Mary-Ann Davies,[1,2] Landon Myer,[5] Jonathan Euvrard ,[1,2] Max Kroon,[6,7] Greg Petro,[8,9] Karen Fieggen,[10,11] Chantal Stewart,[8,12] Natasha Rhoda,[6,7] Stefan Gebhardt,[13,14] Ayesha Osman,[8,15] Kim Anderson,[1] Andrew Boulle,[1,2] Ushma Mehta[1]

AB and UM are joint senior authors.

For numbered affiliations see end of article.

**Correspondence to**
Dr Emma Kalk;
emma.kalk@uct.ac.za

## ABSTRACT

**Purpose** The Western Cape Pregnancy Exposure Registry (PER) was established at two public sector healthcare sentinel sites in the Western Cape province, South Africa, to provide ongoing surveillance of drug exposures in pregnancy and associations with pregnancy outcomes.

**Participants** Established in 2016, all women attending their first antenatal visit at primary care obstetric facilities were enrolled and followed to pregnancy outcome regardless of the site (ie, primary, secondary, tertiary facility). Routine operational obstetric and medical data are digitised from the clinical stationery at the healthcare facilities. Data collection has been integrated into existing services and information platforms and supports routine operations. The PER is situated within the Provincial Health Data Centre, an information exchange that harmonises and consolidates all health-related electronic data in the province. Data are contributed via linkage across a unique identifier. This relationship limits the missing data in the PER, allows validation and avoids misclassification in the population-level data set.

**Findings to date** Approximately 5000 and 3500 pregnant women enter the data set annually at the urban and rural sites, respectively. As of August 2021, >30 000 pregnancies have been recorded and outcomes have been determined for 93%. Analysis of key obstetric and neonatal health indicators derived from the PER are consistent with the aggregate data in the District Health Information System.

**Future plans** This represents significant infrastructure, able to address clinical and epidemiological concerns in a low/middle-income setting.

## INTRODUCTION

Assessing medicine and vaccine safety in pregnancy requires ongoing surveillance across multiple settings. In high-income countries, reviews of outpatient prescriptions and self-medication during pregnancy estimated exposure rates of up to 93% and 43%, respectively, excluding vitamins and supplements.[1 2] Reports from Africa, the site of mass prevention and treatment campaigns for HIV, tuberculosis and malaria, are less frequent: we estimate that 79%–99% of women in Cape Town use medicines antenatally.[3]

Pregnant women have been systematically excluded from pharmaceutical trials and the efficacy, dosing and safety of many medicines used during pregnancy are uncertain or findings are delayed until after the product is licensed and in use. Post-authorisation safety assessments have historically relied on passive reporting of suspected medicine-related adverse events. Such systems have been limited by their dependence on voluntary reporting, variable data quality, absence of background rates of adverse birth outcomes including common congenital disorders, and lack of data to establish a denominator.

Recently, pharmacovigilance in pregnancy has drawn public and political attention following concerns about the initial signal of potential association observed between the

---

**STRENGTHS AND LIMITATIONS OF THIS STUDY**

⇒ The Western Cape Pregnancy Exposure Registry (PER) provides ongoing surveillance of drug exposures in pregnancy and associations with pregnancy outcomes.

⇒ Data collection is integrated into existing services and information platforms and supports routine operations.

⇒ The PER is situated within the Provincial Health Data Centre, an information exchange that harmonises and consolidates all health-related electronic data which limits missing data, allows validation and avoids misclassification in the population-level data set.

⇒ The PER digitises clinical data recorded in operational stationery and we cannot control for data quality nor account for missing data in the source documents nor for unmeasured confounders.

⇒ Medicines obtained outside the public sector systems and traditional and complementary medicines are not included unless they are documented in the clinical stationery.

antiretroviral integrase inhibitor, dolutegravir, and neural tube defects,[4] [5] the potential risk of isoniazid preventive therapy in women living with HIV[6] (WLHIV), and SARS-CoV-2 vaccines.[7] With all these exposures, synthesis and meta-analysis of the available data has been re-assuring and the WHO guidelines recommends no contraindication to their use in pregnant and breastfeeding women.[8–10] In addition, there have been increased calls globally for the inclusion of pregnant women in clinical trials for new therapeutic and preventive agents, particularly in the field of infectious disease.[11–14]

Pregnancy Exposure Registries (PER) are a form of surveillance, designed to iteratively detect adverse events within a defined pregnant population. Importantly, the prospective nature of PER allows collection of exposure and other data before the pregnancy outcome is known. The pharmaceutical industry maintains drug-specific registries for medicines and/or drug classes with known/suspected teratogenic effects (eg, Anti-Epileptic Drug Registries) or as part of post-marketing commitments (eg, the Antiretroviral Pregnancy Register).[15] [16] In addition, teratology information services may collect data on pregnancy exposures. These PER depend on voluntary enrolment by clinicians and/or women, and many do not directly collect data from comparator groups but rely either on internal comparators or on an identified external comparator to provide background prevalence data.[17] Background rates of adverse maternal and obstetric outcomes are necessary to determine deviations from expected proportions (signals). Such data may be limited or lacking in low/middle income countries[18] [19] or differ sufficiently from the source population so as to introduce bias (eg, use of the Metropolitan Atlanta Congenital Defects Program as external comparator for USA-based studies.[17])

The UNICEF/UNDP/World Bank/WHO Special Programme for Research & Training in Tropical Diseases (TDR) has developed a PER approach for resource-limited settings aimed at prospective data collection on exposures in a cohort of pregnant women attending antenatal care services at sentinel sites. Important for validity and causality determination, the approach recommends inclusion of *all* women presenting to the site to allow concurrent establishment of background rates and assessment of multiple potential exposures.[20]

The Western Cape (WC) PER was established in Cape Town in 2016, adapted from the TDR template. It was nested within the province-wide health information exchange, a component within a larger project designed to assess the impact of WHO Option B+ for vertical HIV transmission prevention (ie, universal lifelong antiretroviral therapy (ART) for pregnant and breastfeeding women) at the population and individual levels.[21] Situating the PER within the linked information exchange avoided some of the limitations of exclusive primary care databases in that both electronic inpatient and outpatient prescriptions are recorded as well as those from specialist and other off-site clinics, sources which may be absent from primary care records.[3] [22] [23] The design also supports augmentation of the electronic clinical record for enrolled women, while providing a more secure, sustainable and ethically-viable platform for capturing clinical data on mothers and infants.

We took a pragmatic approach to the establishment of the PER based on the availability of resources and the desire to integrate into existing systems and operational routines, avoiding a parallel infrastructure and supporting longevity. Data generated by the initiative are available for the evaluation and improvement of clinical care as well as epidemiological review.

## COHORT DESCRIPTION

The PER has been established at two sentinel sites in the WC. Gugulethu Midwife Obstetrics Unit (GMOU) provides obstetric care to approximately 5000 women annually in Gugulethu, Cape Town a low-income area with high unemployment and an antenatal HIV prevalence of approximately 30%. GMOU refers patients to Mowbray Maternity Hospital (secondary) and Groote Schuur Hospital (tertiary). About half of all women who attend GMOU are referred to hospital, antenatally or perinatally. Worcester MOU (WMOU) is situated adjacent to the Worcester Provincial Hospital in Worcester, a town of approximately 230 000 in a farming community 120 km outside Cape Town. WMOU provides delivery services for ~3600 women annually. The antenatal HIV prevalence is approximately 16%. Women requiring more advanced care are referred to Worcester Hospital (secondary) and Tygerberg Hospital (tertiary). The community is structurally disadvantaged, and many depend on seasonal employment on farms. In both areas the population is mobile; women move within the WC province and may deliver outside the proscribed referral axes.

Enrolment started at GMOU in Cape Town in September 2016 and at WMOU in January 2018.

All women seeking care at the sentinel primary care sites were included. Most women who use public maternity services, including those with medical and obstetric complications, initially present to primary care, therefore **situating enrolment at the primary care facility allowed us to capture a sample representative of the pregnant population in the geographical drainage area of that facility**.

### Maternal and child health services in the Western Cape

Obstetric care is free at the point of service and approximately 65% of women present at/before 20 weeks gestation.[24] Antenatal care for uncomplicated pregnancies is provided at Basic Antenatal Clinics and MOUs, the latter able to manage uncomplicated vaginal deliveries. At any stage during pregnancy or peripartum women can be referred to district, regional or tertiary hospitals according to standard operating procedures. HIV testing is routine at time points throughout gestation and WLHIV are initiated/re-initiated on ART[25]; those already receiving ART may transfer their HIV care to the MOU. Clients with

other underlying medical conditions (eg, pre-existing hypertension, diabetes mellitus, cardiac conditions) and/or who develop pregnancy-related medical conditions (eg, hypertensive disorders of pregnancy, gestational diabetes) continue antenatal care at hospital. The MOU dispenses ART and antenatal supplements and preventive therapies recommended by the WHO in pregnancy (ie, iron and folate supplements, tetanus and influenza vaccines).[26] Midwives treat the common complaints of pregnancy (heartburn, nausea), urinary tract infection, vaginal candidiasis; and provide syndromic treatment for sexually transmitted infections (STI). Frequently, these medicines are dispensed directly from *ward-stock* without a linked digital record, although details are recorded in paper-based registers.

Within resource constraints, the WC endeavours to provide an antenatal ultrasound scan to clients before 22 weeks gestation for determining gestational age. If concerns are identified women are up-referred for formal fetal anomaly review.

Antenatal visits, HIV testing, transfers and deliveries are recorded against patient names in individual paper-based registers. Monthly aggregate statistics of key obstetric indicators (table 1) are manually counted from these registers and submitted centrally as part of the routine District Health Information System platform.

### Follow-up

The Maternity Case Record (MCR) is a patient-held paper-based document distributed at the first antenatal visit that serves as a record of all clinical obstetric care until discharge after pregnancy outcome, regardless of level of care. It is used throughout South Africa and archived at the site of outcome. Chronic medication and any agents dispensed during pregnancy should be recorded in the MCR by the attending clinicians. However, medicines received at specialist clinics, during hospital admissions and over-the-counter medicines are often not documented.[3 22]

After birth, live- and stillborn neonates are examined by the attending clinician (nurse midwife/doctor) and the outcome of the limited neonatal surface examination is recorded in the MCR. This examination has been shown to detect most major congenital malformations in neonates, that is, those that are visible and do not require diagnostic tools.[27] At GMOU, a clinician employed by the PER performs a review of clinical records to obtain additional data for congenital disorders and stillbirths. In the case of stillbirth, the placenta may be sent for histological examination.

In the WC, most women (99%) give birth at a health facility.[24] Those who do not, will bring their infants to the MOU soon after birth for review and registration.

**For the purposes of the PER, the MCR serves as the primary source of prospectively-collected clinical data.** Thus, women enter the cohort on first visit to the MOU and are followed up until pregnancy outcome.

### Data collection

The PER digitises routinely collected data from the clinical stationery if not already digitised under existing service delivery. In addition to the patient-held MCR, data sources include primary care dating ultrasound reports, and the STI and labour ward delivery registers. As we are using operational data, definitions have been aligned with operational clinical definitions in the WC. Using other routinely collected data elements (gestational age, neonate anthropometry) we are able to align case definitions with those of the Global Alignment of Immunization Safety Assessment in Pregnancy,[28] allowing for harmonisation of data and meaningful comparisons with equivalent data sets. Additionally, we collect or calculate health indicators for the routine monthly aggregate reports required by the MOUs (table 1).

Externally-funded PER data clerks are embedded at the facilities and project-augmented data collection is accommodated within the routine patient and document flow without disruption of clinical care.

The provincial government of the WC operates as a single provider of public sector health services. A 9-digit numeric folder number which is common across the health platform for a given patient facilitates the harmonisation of all electronic health records within the Provincial Health Data Centre (PHDC), the information exchange that consolidates all electronic administrative, pharmacy, laboratory, and disease-specific information.[21] PER data are recorded against this identifier and contribute to the PHDC.

All MOUs use the Primary Healthcare Information Service (PHCIS) electronic medical records system which records attendance against patient identifiers, and ART in WLHIV. PHCIS automatically generates a unique folder number for live infants at birth, providing electronic linkage between mother and baby. Clinicom performs this function at all hospitals. Data are imported daily by the PHDC.[21]

### Completeness of medicine exposure data

Electronic dispensing data in the PHDC are augmented by the PER which captures medicine exposures elicited from the women during the clinical consultation and ward-stock medicines recorded by clinicians in the MCR. The PER also records some lifestyle factors (weight gain, alcohol, tobacco, recreational drugs) that may act as confounders for certain outcomes. Combining the electronic pharmacy data in the PHDC strengthens the ascertainment of exposures, providing a complete list of medication dispensed from public sector pharmacies. Using multiple data sources for this has been shown to provide a more complete picture of antenatal medicine use essential for pregnancy exposure research.[3 29 30]

### Outcome ascertainment

Information on neonatal outcomes such as vital status, birth weight, gestational age and APGAR scores tend to be consistently captured across the cohort. The key

**Table 1** Summary of data elements in the Pregnancy Exposure Registry

| Variables collected | Calculated parameters | MOU aggregate statistics |
|---|---|---|
| *Antenatal*<br>Maternal date of birth<br>Date of first antenatal visit<br>Last menstrual period<br>Parity, gravidity<br>Obstetric and medical history<br>Chronic medication<br>Height, mid-upper arm circumference, weight, blood pressure, urinalysis<br>Symphysis fundal height<br>Alcohol, tobacco, drug use<br>Number of antenatal visits | Gestational age at first antenatal visit | Number of first visits<br>Number of women first ANC<20 weeks<br>Number age <20 years or >38 years<br>Number grand multipara (≥5 deliveries)<br>Number high blood pressure/proteinuria |
| *Vertical transmission of HIV*<br>HIV status at first antenatal visit<br>Subsequent positive HIV test<br>HIV treatment incl. regimen switches<br>CD4 count<br>Viral load<br>HIV-exposed infant HIV-PCR | Number of women at high risk of vertical HIVtransmission<br>ART in hand at estimated time of conception<br>ART in hand at delivery | Number of women living with HIV:<br>Before pregnancy<br>During pregnancy<br>Number of women on ART (first and second line):<br>Before, during pregnancy<br>Viral load unsuppressed at pregnancy and delivery<br>Number of infant birth HIV-PCR |
| *Ultrasound*<br>Gestational age<br>Abnormalities<br>Expected date of delivery | | Number of ultrasounds conducted<br>Number multiple pregnancies |
| *Maternal outcome*<br>Facility-based death | Vital status | Maternal death |
| *Peripartum*<br>Date and site of outcome<br>Method of delivery<br>Gestational age at outcome | Prematurity (<37 completed weeks gestation) | Number of deliveries |
| *Pregnancy outcome*<br>Live birth<br>Stillbirth<br>Miscarriage<br>Termination of pregnancy<br>Ectopic pregnancy<br>Molar pregnancy | Gestational age at pregnancy outcome | Number of livebirths, stillbirths, miscarriages |
| *Neonate*<br>Date of birth<br>Sex, APGAR scores<br>Gestational age<br>Birth weight, length, head circumference, foot length<br>Neonatal surface examination<br>Abnormalities noted | Gestational age at birth<br>Low birth weight (<2500 g)<br>Prematurity (<37 completed weeks gestation)<br>Neonatal death | Number of low birth weight infants<br>Number premature infants<br>Number neonatal deaths<br>Perinatal mortality rate |

ANC, antenatal care; ART, antiretroviral therapy; MOU, midwife obstetric unit.

findings of the neonatal surface examinations, although often perfunctory, usually result in the recording of notable physical anomalies. Internal anomalies such as cleft palate, hip dysplasias and cardiac anomalies as well as more subtle dysmorphic features may be missed at the time of the initial neonatal examination. Details of neonatal deaths, and major congenital disorders often require review of inpatient records at the delivery facilities.

PER data are imported daily into the PHDC and linked using patient identifiers, providing a comprehensive electronic clinical record at the level of the individual which is accessible to the attending clinicians. Both systems benefit greatly from this design. The PER allows

for validation of the provincial data set as relates to pregnancy and delivery, and the PHDC is able to identify missing outcomes (often at sites outside the referral axes) or exposures (from electronic pharmacy dispensing) not included in the PER.

## FINDINGS TO DATE

Between 01 September 2016 and 31 August 2021, 31 346 pregnancies were recorded in the PER. To assess robustness of the data set, we analysed data for a subset of women who attended their first visit to antenatal care between 01 January 2018 and 31 December 2019 (table 2). Over this 2-year period, 14 527 individual pregnancies were recorded in the PER: 9435 and 5092 at the urban and rural site, respectively. Outcomes were determined for 93.4% of pregnancies (n=13 574). Gestational dating scans were performed in 38.5% (n=5583) of all enrolees, of whom 60% (n=3345) were ≤22 weeks, facilitating more precise gestational dating at birth as well as timing of exposures. Overall, 1287 women (9%) were exposed to medicines with pregnancy safety surveillance requirements (table 3 and online supplemental table 1).

Where relevant, we compared rates of key adverse birth outcomes in the PER with official aggregate routine indicator data for the WC,[24 31–33] derived from register aggregates reported through the District Health Information System (DHIS) (table 3). The comparisons are re-assuring across both the urban and rural sites, validating the indicator outputs of the PER and demonstrating utility to the services. The data will contribute to detailed aggregate reports for facility managers and streamline the monthly submissions to the DHIS which are currently based on manual counts.

The congenital disorder data are still being cleaned for analysis with pregnancy outcomes.

## Published and other outputs

We conducted an initial baseline assessment comparing clinical records to dispensing data before the implementation of the PER[22] and recently updated the analysis demonstrating the value of combining PER and electronic pharmacy data in improving medicine exposure ascertainment.[3] We are currently investigating the impact of data source on gestational age (Malaba T, manuscript in preparation) and hypertensive disorders of pregnancy.[34] PER data have contributed to population-based analyses describing the use and safety of sodium valproate and isoniazid for tuberculosis preventive therapy in pregnancy.[35 36] In addition, initiation of the PER provided the opportunity to host a workshop, *Building Teratovigilance Capacity in Africa,* which provided networking and training opportunities to 60 delegates from sub-Saharan Africa https://globalpharmacovigilance.tghn.org/resources/building-teratovigilance-capacity-africa/.

## System strengthening

In addition to the employment of project-specific staff, embedded with computers at the facilities, the project supports ongoing training of clinical staff to improve and standardise clinical history-taking with an emphasis on exposures, neonatal examination and clinical record keeping. Open resources include the WHO/TDR *Stepwise Surface Examination of the Newborn* (https://www.who.int/tdr/publications/videos/stepwise-surface-examination-newborns/en/) and the training modules for midwives we developed as part of the South African Central Pregnancy Exposure Registry (https://www.ubomibuhle.org.za/training-lessons).[37] These resources are freely available and are now in use at PER sites across South Africa.

## STRENGTHS AND WEAKNESSES
### Strengths

The integration of the PER within the PHDC greatly increases the completeness of the data. It facilitates identification of pregnancy outcomes at facilities outside our sentinel referral chains reducing loss to follow-up. Harmonisation and triangulation of two data sources for medicine exposures (ie, clinical records and electronic pharmacy records) provides a more robust summary of exposures than either alone.[3 17 22] These systems comprise unique infrastructure able to address clinical and public health concerns in a low/middle-income setting.

Accurate timing of exposures over the course of pregnancy is crucial to assess potential associations with adverse pregnancy outcomes. Collecting multiple reference points for gestational age (ie, neonatal record, ultrasound, last menstrual period, symphysis-fundal height) enabled the development of a hierarchy of methods and the allocation of a confidence score to the reported gestational age.[38–40] This offers an advantage over insurance claims data sets which are often used to determine safety information and in which pregnancy and gestational age must be inferred from clinical coding alone.

In line with the TDR protocol,[20] all women attending the PER primary care sites are enrolled and we reflect background rates of important pregnancy parameters similar to what is expected from national aggregate data. This will be expanded to include background rates for congenital disorders, data which are lacking in South Africa.[41] This structure also allows for the analysis of multiple current and potential future exposures and emerging health concerns, for example, novel medicines and vaccines such as for SARS-CoV-2. Determining the rates and associations of rare events such as major congenital anomalies requires large, representative samples. Such analyses necessitate resources for data cleaning and interpretation, especially to determine the timing of drug/teratogen exposures over the course of gestation. This work is currently underway in the PER.

From the outset, it was important to avoid a parallel system and support project sustainability. The PER has been integrated into the existing clinical and clerical

**Table 2** Maternal and obstetrical characteristics of the cohort 2018–2019

| Variable | PER total<br>n=14 527 | PER urban<br>n=9435 (65%) | PER rural<br>n=5092 (35%) |
|---|---|---|---|
| Age (years) median (IQR) | 27 (23–32) | 28 (23–33) | 26 (22–31) |
| Living with HIV at pregnancy outcome | 3931 (27.1) | 3241 (34.3) | 690 (13.6) |
| Obstetric ultrasound present n (%) | 5583 (38.4) | 4063 (43.1) | 1520 (29.9) |
| Early ultrasound (ie, <22 weeks) n (% of ultrasound) | 3345 (59.9) | 2393 (58.9) | 952 (62.6) |
| Potentially unsafe medicine exposure | 1287 (9.0) | 857 (9.3) | 430 (8.5) |
| Gestational age at birth (weeks) median (IQR) | 40 (37–40) | 40 (36–40) | 39 (35–40) |
| Birth weight (g) median (IQR) | 3100 (2750–3440) | 3140 (2800–3480) | 2975 (2575–3320) |
| Low birth weight* n (%) | 1736 (12.0) | 879 (9.3) | 857 (16.8) |
| Premature birth† n(%) | 2949 (20.3) | 1735 (18.4) | 1214 (23.8) |
| Pregnancy outcome n (%) | | | |
| Live birth | 12 419 (85.5) | 1189 (82.3) | 4630 (90.9) |
| Stillbirth | 296 (2.0) | 180 (1.9) | 116 (2.3) |
| Neonatal death‡ | 109 (0.8) | 71 (0.5) | 36 (0.7) |
| Miscarriage | 395 (2.7) | 318 (3.4) | 77 (1.5) |
| Ectopic pregnancy | 82 (0.6) | 60 (0.6) | 22 (0.4) |
| Termination of pregnancy | 273 (1.9) | 223 (2.4) | 50 (1.0) |
| Unknown | 953 (6.6) | 792 (8.4) | 161 (3.1) |
| Delivery method§ n(%) | | | |
| Born before arrival at birthing facility | 608 (4.7) | 245 (3.1) | 363 (7.6) |
| Vaginal delivery | 7587 (59.2) | 4655 (57.9) | 2932 (61.3) |
| Assisted delivery¶ | 140 (1.1) | 51 (0.6) | 89 (1.9) |
| Caesarean section | 3416 (26.6) | 2411 (30.0) | 1005 (21.0) |
| Unknown | 1073 (8.4) | 680 (8.5) | 393 (8.2) |
| Infant outcome§ n(%) | | | |
| Stillborn | 296 (2.3) | 180 (2.2) | 116 (2.4) |
| Early neonatal death‡ | 80 (0.6) | 55 (0.7) | 25 (0.5) |
| Late neonatal death | 29 (0.2) | 18 (0.2) | 11 (0.2) |
| Alive | 12 419 (96.8) | 7798 (96.9) | 4630 (96.8) |
| Tobacco use** n(%) | | | |
| Current user | 1297 (8.9) | 87 (0.9) | 1210 (23.8) |
| Past user | 55 (0.4) | 13 (0.1) | 42 (0.8) |
| Never user | 9997 (68.8) | 7222 (76.5) | 2775 (54.5) |
| Not reported | 3178 (21.9) | 2113 (14.5) | 1065 (7.3) |
| Alcohol use** n(%) | | | |
| Current user | 588 (4.1) | 339 (3.6) | 249 (4.9) |
| Past user | 167 (1.2) | 66 (0.7) | 101 (2.0) |
| Never user | 10 570 (72/8) | 6885 (73.0) | 3685 (72.4) |
| Not reported | 3202 (22.0) | 2145 (14.8) | 1057 (7.3) |

*Birthweight <2500 g; liveborn infants only.
†Birth <37 completed weeks gestation; liveborn infants only.
‡Neonatal death: death before 28 days of life; early neonatal death: death before 7 days of life; late neonatal death: death between 8 and 28 days of life.
§Viable pregnancies (ie, >27 weeks gestation) (n=12 824).
¶Forceps or vacuum delivery.
**Reported at first antenatal visit.
PER, Pregnancy Exposure Registry.

**Table 3** Comparison between reported or calculated PER outcomes and aggregate indicators in formal provincial information systems

| Indicator | PER total n=14 527 | PER urban n = 9435 (65%) | PER rural n=5092 (35%) | Reported aggregate for the Western Cape 2017–2019* |
|---|---|---|---|---|
| Stillbirth† n (%) | 296 (2.0) | 180 (1.9) | 116 (2.4) | 2.2%[33] |
| Per 1000 births | 20 | 19.1 | 24 | 18.531 |
| | | | | 22.1[31 32] |
| Neonatal death in facility rate‡ per 1000 live births | 8.7 | 9.2 | 7.7 | 8.9[31 32] |
| Perinatal mortality rate§ per 1000 births | 29 | 29 | 29 | 25.6[31] |
| | | | | 27.9[32] |
| | | | | 29.1[33] |
| Low birth weight¶ n(%) | 1737 (12.0) | 879 (9.3) | 857 (16.8) | 14.9% urban subdistrict |
| | | | | 18.4% rural subdistrict[33] |
| Maternal mortality in facility ratio per 100 000 live births | | 63.5 | Insufficient data | 43.6–66.8[32] |
| Teenage pregnancies (10–19 years) n(%) | 929 (6.4) | 450 (4.8) | 497 (9.4) | 3.5% urban subdistrict |
| | | | | 7.3% rural subdistrict[33] |
| Caesarean section rate per 1000 births | 3416 (26.6) | 2411 (30.0) | 1005 (21.0) | 28.9[32]–29.3[33] |

*Includes aggregate reports compiled from the District Health Information System and Perinatal Problem Identification Programme.[31–33]
†Delivery of a baby with no signs of life after 27 completed weeks of gestation (ie, viable baby born dead).
‡Death before 28 days of life.
§Stillbirth plus neonatal deaths <8 days per 1000 births.
¶Birthweight <2500 g; liveborn infants only.
PER, Pregnancy Exposure Registry.

routines and uses local electronic health information platforms. It allows for electronic generation of key monthly indicators at primary care sites that are otherwise collected by hand.

As the cohort expands, capacity to conduct nested studies that facilitate signal detection and signal verification of potential or suspected teratogens will improve. The collection of individual-level data in a large prospectively enrolled cohort, representative of both urban and rural WC populations who use public sector services will support more robust analyses that can better account for confounding factors in such observational data.

### Weaknesses
The PER digitises clinical data recorded in operational stationery and we cannot control for data quality nor account for missing data in the source documents, including the risk of under-reporting. To address this, we have engaged in ongoing training at the sites with an emphasis on drug history-taking, medical record-keeping and neonatal examination offering in-person teaching and video tutorials. Clinical staff have been provided with Medicine Identification Aids with photographs of common formulations and packaging, and the WHO Birth Defects Atlas.[42] However, misclassification remains a potential risk.

Notwithstanding the advantages of the individual-level data available within the PHDC, data are limited to those that are entered into one of the electronic medical records systems used in the public sector. In terms of medicine exposures, the PER documents dispensed medication which may not reflect actual use. In addition, medicines obtained outside of the public sector systems, from private doctors or over-the-counter from pharmacies are not included unless they are noted in the clinical records.[3] Similarly, traditional and complementary medicines lack a linked electronic footprint and are not included.

The PER database is parsimonious by design and necessity and we are unable to account for unmeasured confounders. However, data fields are collected for the entire cohort who are all drawn from the same geographical areas served by the primary care clinics. Additionally, we record limited data on lifestyle factors relevant in pregnancy (weight gain, exposure to tobacco, alcohol, recreational drugs) which are lacking from equivalent population data sets based on insurance claims data.

### COLLABORATION
As with the PHDC within which it is located, the PER can address clinical, operational and research needs, and data access is specific to each. Aggregate reports are

available to managers. Data are anonymised using standard protocols for de-identifying records before they are shared with researchers who are not directly engaged in the women's clinical care. It is anticipated that such de-identified individual-level data may be shared as part of the South African Central Pregnancy Registry[37] and with similar PER initiatives regionally or internationally.[43] Data-sharing commitments are particularly relevant to research of rare events such as congenital disorders.[20] The PHDC has in-built privacy systems and strict governance structures managing the protection and use of health data for both service and research purposes and these apply to the PER.[21]

## Patient and public involvement

The PER is integrated into the data collection and curation services of the Western Cape Government Department of Health and clinical and other service providers have engaged with the project since its inception. The data are available to managers as aggregate reports and to contribute to the electronic clinical records accessible by clinicians. Feedback from users contributes iteratively to optimisation of the PER to improve health outcomes for pregnant women and infants.

## CONCLUSIONS

Research on medicine safety in pregnancy requires data on individual pregnancies, mother–infant linkage, medication exposure, gestational age at exposure and maternal and birth outcomes. Data completeness and robustness continues to improve with ongoing training, evolution of routine clinical information systems and increasing political focus on pregnancy exposures. The cohort is well-placed to detect large signals in pregnancy outcomes as novel maternal exposures are introduced, and to contribute to cohort harmonisation for rarer outcomes and address the lack of information on congenital disorders in Africa.

**Author affiliations**
[1]Centre for Infectious Disease Epidemiology & Research, University of Cape Town Faculty of Health Sciences, Cape Town, South Africa
[2]Health Intelligence Directorate, Western Cape Department of Health, Cape Town, South Africa
[3]Ukwanda Centre for Rural Health, Department of Global Health, Stellenbosch University, Stellenbosch, South Africa
[4]Department of Paediatrics & Child Health, Stellenbosch University, Stellenbosch, South Africa
[5]Division of Epidemiology & Biostatistics, University of Cape Town Faculty of Health Sciences, Cape Town, South Africa
[6]Department of Paediatrics & Child Health, University of Cape Town Faculty of Health Sciences, Cape Town, South Africa
[7]Neonatal Services, Mowbray Maternity Hospital, Cape Town, South Africa
[8]Department of Obstetrics & Gynaecology, University of Cape Town Faculty of Health Sciences, Cape Town, South Africa
[9]Maternity Services, New Somerset Hospital, Cape Town, South Africa
[10]Division of Human Genetics, University of Cape Town Faculty of Health Sciences, Cape Town, South Africa
[11]Medical Genetics Services, Groote Schuur Hospital, Cape Town, South Africa
[12]Maternity Services, Mowbray Maternity Hospital, Cape Town, South Africa
[13]Department of Obstetrics & Gynaecology, Stellenbosch University, Stellenbosch, South Africa
[14]Maternity Services, Tygerberg Hospital, Cape Town, South Africa
[15]Maternity Services, Groote Schuur Hospital, Cape Town, South Africa

**Acknowledgements** We acknowledge and thank the PER data teams in Cape Town and Worcester, and the clinical and management staff at the facilities. The support and investment of the subdistrict and provincial health management is crucial to maintaining the initiative. We further acknowledge colleagues from the Western Cape Government: Health and the City of Cape Town.

**Contributors** Conception and design: EK, UM, AB, ALS, M-AD and LM. Design and implementation including data systems: EK, UM, ALS, GP, KF and JE. Data harmonisation: AH, FP, JE, AB, M-AD and EK. Clinical oversight: GP, MK, CS, NR, SG, AO and KF. Data cleaning and analysis: EK, ALS, UM, KA and AH. All authors critically reviewed the manuscript. EK and AB are guarantors of the pubilcation.

**Funding** This work was supported by the US National Institutes of Health (R01HD080465, U01AI069911, U01AI069924, K43TW010683), Bill and Melinda Gates Foundation (INV-004508; 1164272; 1191327; INV-004657), the Wellcome Trust (203135/Z/16/Z), the International AIDS Society (2017-518/SLO), the United States Agency for International Development (72067418CA00023) and the US Centres for Disease Control and Prevention (GH001934).

**Competing interests** EK, AB, M-AD and KA received funding from Viiv Healthcare unrelated to this project.

**Patient and public involvement** Patients and/or the public were not involved in the design, or conduct, or reporting, or dissemination plans of this research.

**Patient consent for publication** Not applicable.

**Ethics approval** This study involves human participants and was approved by Faculty of Health Sciences Human Research Ethics Committees of the University of Cape Town: HREC: 749/2015. Faculty of Health Sciences Human Research Ethics Committees of Stellenbosch University: N17/04/040 and N20/08/084. The PER involves digitisation of routine operational data using the software and data systems of the Western Cape Government Department of Health. Data are stored and curated by the Western Cape Government Department of Health and fall within its legal and ethical protections. Access is strictly governed. Anonymised data sets are provided to researchers after protocol and ethical review by the Department and the University IRB. A waiver of informed consent was granted by the institutions concerned.

**Provenance and peer review** Not commissioned; externally peer reviewed.

**Data availability statement** Data may be obtained from a third party and are not publicly available.

**ORCID iDs**
Emma Kalk http://orcid.org/0000-0001-7706-6866
Jonathan Euvrard http://orcid.org/0000-0002-3937-7855

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
