## [Reviewer comments · BMJ Open]

ARTICLE DETAILS

TITLE (PROVISIONAL)	Cohort Profile: The Western Cape Pregnancy Exposure Registry (WCPER)
AUTHORS	Kalk, Emma; Heekes, Alexa; Slogrove, Amy; Phelanyane, Florence; Davies, Mary-Ann; Myer, Landon; Euvrard, Jonathan; Kroon, Max; Petro, Greg; Fieggen, Karen; Stewart, Chantal; Rhoda, Natasha; Gebhardt, Stefan; Osman, Ayesha; Anderson, Kim; Boulle, Andrew; Mehta, Ushma

VERSION 1 – REVIEW

REVIEWER	Renaud, Francoise World Health Organization, HIV Hep STI
REVIEW RETURNED	03-Feb-2022

GENERAL COMMENTS	General comments:  - The paper reports on a very useful and innovative work that is the use and linkages of eHealth data for improving the surveillance of drugs in pregnancy – this program will serve for extending surveillance in other countries of the region. Having a paper published and reporting on operations, progress and way forward documents a public health approach. - The paper shows that the program covers a large sample of data with a capacity to contribute to regional or global reports for continuous surveillance as well as the detection of rare outcomes as new drugs are introduced. Notably, WHO is interested in such programs to be ready and operational as new ARV are introduced, such as long-acting formulations for treatment or Prep. - The objectives of the paper to describe, take stock and report on strengths and weaknesses, is well defined. It shows are the foundations of the infrastructure are robust – linking patient monitoring systems that are operational and hosted in a central structure. - the reviewer encourages the authors to better document the efforts provided at the sites for supporting data collection (not only data entry), strengthening data quality (an issue is raised in the abstract, but the response is not entirely satisfactory), and ensuring data and outcome harmonization and standardization across studies/countries/global initiatives. This information will serve similar initiatives to take off. - Some examples such as the standardization of neonate surface exam, ensuring all neonates is examined, training of midwives, investment in HR would be illustrative. - Such initiative should be encouraged and sustained, and regular reporting/publishing is encouraged. - Looking at the future: for the longevity of the manuscript and program, the reviewer invite the authors to contextualize the program into a global framework for accelerated inclusion of pregnant women in pre-licensure clinical trials where surveillance is a key piece – see WHO/IMPAACT/IAS meeting report at: https://www.who.int/publications/i/item/9789240040182 Detailed comments:
--

	- Intro: Line 28: on dosing and safety of many medicines used during pregnancy are “uncertain” – suggest replacing by data on dosing and safety are delayed as most often obtained once the drug is introduced, especially on rare pregnancy outcomes that require large samples of women receiving the drug. - Line 33: use “have been “ as there are a number of disease registries, drug registries and active surveillance programs established that are addressing the issue - ref the inventory and report of PREGLAC - Line 43. This paragraph is misleading as it could be interpreted that NTD signal is still as high as in 2018 and 2019. a) The reviewer requests to update the text and references using Zash 2020/2021 papers and WHO 2021 ARV Guidelines that shows a significant decrease in trends and WHO recommends the use of DTG in all populations including pregnant women (BOX 4.7. at: https://www.who.int/publications/i/item/9789240031593) and b) similarly on the use of IPT in pregnancy, a recent meta-analysis was conducted that the authors are encouraged to consider to clarify that there is no contra-indication for use in pregnancy as per WHO recommendations (at: 1.4. Tuberculosis preventive treatment options TB Knowledge Sharing (tbksp.org) and bottom of p 7 of https://www.who.int/tb/publications/2017/tb_guidelines2017_annex6_en_v4.pdf); for vaccines against COVID 19, the authors could consider adding that it is also an encouraging example on how surveillance in 2021 has been rapidly implemented and generating safety data in pregnancy in real time. - Lines 29 and 44 : Replace WHO by “ The UNICEF/UNDP/World Bank/WHO Special Programme for Research & Training in Tropical” (TDR) and TDR
--	---

REVIEWER	Schueler-Faccini, Lavinia Univ Fed Rio Grande do Sul
REVIEW RETURNED	06-Feb-2022

GENERAL COMMENTS	This is a very important manuscript since it presents an impressive registry in a middle-income country, one of the BRICS, addressing maternal-infant health. Although high-income countries have sound data on exposures during pregnancy and related outcomes, the same is not true for low and middle-income countries, especially considering maternal exposures during pregnancy and congenital anomalies. Saying so, I have a major comment for the authors: In the abstract, it is stated that "(PER) was established ... to provide on-going surveillance of drug exposures in pregnancy and associations with pregnancy outcomes." The introduction stresses the importance of the pharmacovigilance for pregnant women and in the methodology the inclusion of congenital anomalies detected and the surface exam are described. However in the results, nothing is mentioned about the use of medications during pregnancy, nor about frequencies or types of abnormalities seen in the babies registered. Although the authors briefly mention that some of their results on pharmacovigilance were published already in this manuscript nothing is presented in the Tables (Tobacco and Alcohol are described, but NOT medications). Therefore, the authors should include some of the findings on medicines exposures and the outcomes of surface exams (birth defects/congenital anomalies), or discuss why they don't present it. Minor Points: 1. Table 1. The acronym MOU should be described in the legends. 2. I'm curious why chronic diseases (e.g. maternal diabetes) are not included in the data collected, as well as infections (at least the STORCH ones) In summary I congratulate the authors for this very important registry and for the effort to present in a comprehensive manuscript.
---

VERSION 1 – AUTHOR RESPONSE

Reviewer 1

	Comment	response
1. 1	The reviewer encourages the authors to better document the efforts provided at the sites for supporting data collection (not only data entry), strengthening data quality (an issue is raised in the abstract, but the response is not entirely satisfactory), and ensuring data and outcome harmonization and standardization across studies/countries/global initiatives. This information will serve similar initiatives to take off. Some examples such as the standardization of neonate surface exam, ensuring all neonates is examined, training of midwives, investment in HR would be illustrative	The following section has been added to the text (in addition to that on page 16) “System strengthening In addition to the employment of project-specific staff, embedded with computers at the facilities, the project supports on-going training of clinical staff to improve and standardize clinical history-taking with an emphasis on exposures, neonatal examination and clinical record keeping. Open resources include the WHO/TDR Stepwise Surface Examination of the Newborn (https://www.who.int/tdr/publications/ideas/stepwise-surface-examination-newborns/en/) and the training modules for midwives we developed as part of the South African National Pregnancy Exposure Registry (https://www.ubomibuhle.org.za/training-lessons)[33]. These resources are freely available and are now in use at PER sites across South Africa.”
1. 2	The reviewer invite the authors to contextualize the program into a global framework for accelerated inclusion of pregnant women in pre-licensure clinical trials where surveillance is a key piece – see WHO/IMPAACT/IAS meeting report at: https://www.who.int/publications/i/item/9789240040182	Thank you. The following text as been added: “In addition, there have been increased calls globally for the inclusion of pregnant women in clinical trials for new therapeutic and preventive agents, particularly in the field of infectious disease[11-14].”
1. 3	- Intro: Line 28: on dosing and safety of many medicines used during pregnancy are “uncertain” – suggest replacing by data on dosing and safety are delayed as most often obtained once the drug is introduced, especially on rare pregnancy outcomes that require large samples of women receiving the drug.	The sentence has been changed to the following: “Pregnant women have been systematically excluded from pharmaceutical trials and the efficacy, dosing and safety of many medicines used during pregnancy are uncertain or findings delayed until after the product is licensed and in use.” Uncertainty remains for many products even after years of use owing to the limitations of post-authorization safety assessments.
1. 4	Line 33: use “have been “ as there are a number of disease registries, drug registries and active surveillance programs established that are addressing the issue - ref the inventory and report of PREGLAC	The clause has been added as recommended.
1. 5	DTG in all populations including pregnant women (BOX 4.7. at: https://www.who.int/publications/i/item/978924003159	The text has been amended to clarify that these concerns have been allayed. Page 5.

	3) and b) similarly on the use of IPT in pregnancy, a recent meta-analysis was conducted that the authors are encouraged to consider to clarify that there is no contra-indication for use in pregnancy as per WHO recommendations (at: 1.4. Tuberculosis preventive treatment options TB Knowledge Sharing (tbksp.org) and bottom of p 7 of https://www.who.int/tb/publications/2017/tb_guidelines2017_annex6_en_v4.pdf); for vaccines against COVID 19, the authors could consider adding that it is also an encouraging example on how surveillance in 2021 has been rapidly implemented and generating safety data in pregnancy in real time.”	“Recently, pharmacovigilance in pregnancy has drawn public and political attention following concerns about the initial association observed between the antiretroviral integrase inhibitor, dolutegavir, and neural tube defects [4, 5], the potential risk of isoniazid preventive therapy in women living with HIV [6] (WLHIV), and SARS-CoV-2 vaccines [7]. With all these exposures, synthesis and meta-analysis of the available data has been re-assuring and the WHO guidelines report no contra-indication to their use in pregnant and breast-feeding women [new references 8 - 10].
1.6	Lines 29 and 44 : Replace WHO by “ The UNICEF/UNDP/World Bank/WHO Special Programme for Research & Training in Tropical Diseases” (TDR) and TDR	The substitutions have been made as recommended.

Reviewer 2

	Comment	Response
2.1	However in the results, nothing is mentioned about the use of medications during pregnancy, nor about frequencies or types of abnormalities seen in the babies registered. Although the authors briefly mention that some of their results on pharmacovigilance were published already in this manuscript nothing is presented in the Tables (Tobacco and Alcohol are described, but NOT medications). Therefore, the authors should include some of the findings on medicines exposures and the outcomes of surface exams (birth defects/congenital anomalies), or discuss why they don't present it.	Thank you, the point is well-taken. A row has been included in Table 2 noting the frequency (and %) of any potentially unsafe medication flagged at any stage over the course of gestation. The timing of exposure is not included in this manuscript. A list of potentially unsafe medication is listed in Supplementary Table 1. In addition, the following text has been added: At the urban site, 38 congenital disorders in live births were confirmed in 2018 – 2019 (Table 4.) Twelve were classified as minor (pre-axial polydactyly, undescended testes, subglottic stenosis not requiring intervention). Major congenital disorders included two cases of fetal hydantoin syndrome (both diagnosed antenatally) and four infants with neural tube defects (two identified antenatally, two at birth). The congenital disorder data are still being cleaned for analysis with pregnancy outcomes.
2.2	Table 1. The acronym MOU should be described in the legends.	Thanks, this has been added.
2.3	I'm curious why chronic diseases (e.g. maternal diabetes) are not included in the data collected, as well as infections (at least the STORCH ones)	The following text has been added: In addition to HIV and TB infections, hypertensive disorders of pregnancy and diabetes mellitus (pre-existing and gestational) are well-defined in the PHDC and can be included in assessments. Other medical conditions can be inferred from prescription data. Syphilis exposure is recorded if documented in the clinical stationery and deduced from record of intramuscular benzyl penicillin dispensing. The other (TORCH) infections will be noted against a patient record if

	the assay is performed in an NHLS laboratory. At present, these data are not presented in the PER but will be at a future iteration.
--	---

VERSION 2 – REVIEW

REVIEWER	Renaud, Francoise World Health Organization, HIV Hep STI
REVIEW RETURNED	19-Apr-2022

GENERAL COMMENTS	The authors have addressed most of the comments, and the paper is improved. Below are remaining comments that need to be completed before publication:  - Page 5 – last paragraph: initial association is wrong, replace by “initial signal of a potential association”. Also, replace “reports no contra-indications” with “recommends”. - Page 6 – 2nd paragraph: in the sentence “The pharmaceutical industry maintains drug-specific registries for medicines and/or drug classes with known/suspected teratogenic effects or as part of post-marketing commitments; e.g., the Antiretroviral Pregnancy and Anti-Epileptic Drug Registries”, the order of examples is confusing. The authors should avoid unnecessary confusion between ART and anti-epileptic drugs. Repace by: “The pharmaceutical industry maintains drug-specific registries for medicines and/or drug classes with known/suspected teratogenic effects (e.g. Anti-Epileptic Drug Registries) or as part of post-marketing commitments (e.g., the Antiretroviral Pregnancy) and - Page 13 – 1st paragraph: the use of “potentially unsafe medicines” is misleading. Replace it with “medicines with active surveillance of safety during pregnancy” – - Page 13, 2nd paragraph:  o the data on congenital anomalies are not validated yet and cannot be properly analyzed as per “The congenital disorder data are still being cleaned for analysis with pregnancy outcomes. “ These data should not be published without stringent analysis and validation and robust interpretation – the sub-section of the paragraph on urban site and congenital disorders should be removed in absence of proper review and background rates, as well as table 4. o The reviewer asks that the authors use the critical example of collecting and validating data on congenital anomalies, especially severe and rare anomalies needing large samples, to illustrate the requirements and remaining work that needs to be brought to the PER, to allow for data cleaning, statistical interpretation, especially in the case of drug exposure, and any other variable such as the timing of exposure. This could be discussed under weaknesses on page 16. o Instead, the authors should elaborate on the comparison conducted and presented in Table 3 – how does the comparison perform and what can they say about the robustness of data collected (quality, quantity, definitions, interoperability of databases, etc...) to explain the “Where relevant” introduction of the paragraph. - Page 15 – 3rd paragraph: Replace “In line with WHO recommendations” with “ In line with the TDR protocol”
---

	- Page 16 – 1st paragraph under weaknesses: in addition to misclassification, consider under-reporting with the difficulty to establish baseline rates - Ref [8] for TB drug: the link is not the right one. it is for TPT in pregnant women with HIV then the latest guidelines are the ones from 2020; WHO position has not changed. Should use: https://tbksp.org/en/node/43
--	--

VERSION 2 – AUTHOR RESPONSE

Reviewer 1

	Comment	response
1.1	Page 5 – last paragraph: initial association is wrong, replace by “initial signal of a potential association”. Also, replace “reports no contra-indications” with “recommends”.	The following changes have been applied, “Recently, pharmacovigilance in pregnancy has drawn public and political attention following concerns about the initial signal of potential association observed between the antiretroviral integrase inhibitor, dolutegavir, and neural tube defects[4, 5], the potential risk of isoniazid preventive therapy in women living with HIV[6] (WLHIV), and SARS-CoV-2 vaccines[7]. With all these exposures, synthesis and meta-analysis of the available data has been re-assuring and the World Health Organization (WHO) guidelines recommends no contra-indication to their use in pregnant and breast-feeding women [8-10].”
1.2	Page 6 – 2nd paragraph: in the sentence “The pharmaceutical industry maintains drug-specific registries for medicines and/or drug classes with known/suspected teratogenic effects or as part of post-marketing commitments; e.g., the Antiretroviral Pregnancy and Anti-Epileptic Drug Registries”, the order of examples is confusing. The authors should avoid unnecessary confusion between ART and anti-epileptic drugs. Repace by: “The pharmaceutical industry maintains drug-specific registries for medicines and/or drug classes with known/suspected teratogenic effects (e.g. Anti-Epileptic Drug Registries) or as part of post-marketing commitments (e.g., the Antiretroviral Pregnancy) and	The following changes have been applied, “The pharmaceutical industry maintains drug-specific registries for medicines and/or drug classes with known/suspected teratogenic effects (e.g., Anti-Epileptic Drug Registries) or as part of post-marketing commitments; (e.g., the Antiretroviral Pregnancy Register) [15, 16].”
1.3	- Page 13 – 1st paragraph: the use of “potentially unsafe medicines” is misleading. Replace it with “medicines with active surveillance of safety during pregnancy” –	The following changes have been applied, “Overall, 1287 women (9%) were exposed to medicines with pregnancy safety surveillance requirements (Table 3 and Supplementary Table 1).”
1.4	o the data on congenital anomalies are not validated yet and cannot be properly analyzed as per “The congenital disorder data are still being cleaned for analysis with pregnancy outcomes. “ These data should not be published without stringent analysis and validation and robust	The comment is well-received and the text and Table 4 have been deleted.

	interpretation – the sub-section of the paragraph on urban site and congenital disorders should be removed in absence of proper review and background rates, as well as table 4.	
1.5	o The reviewer asks that the authors use the critical example of collecting and validating data on congenital anomalies, especially severe and rare anomalies needing large samples, to illustrate the requirements and remaining work that needs to be brought to the PER, to allow for data cleaning, statistical interpretation, especially in the case of drug exposure, and any other variable such as the timing of exposure. This could be discussed under weaknesses on page 16.	Thank you. This is a valid point. The following text has been added (pg 15): “Determining the rates and associations of rare events such as major congenital anomalies requires large, representative samples. Such analyses necessitate resources for data cleaning and interpretation, especially to determine the timing of drug/teratogen exposures over the course of gestation. This work is currently underway in the PER.”
1.6	o Instead, the authors should elaborate on the comparison conducted and presented in Table 3 – how does the comparison perform and what can they say about the robustness of data collected (quality, quantity, definitions, interoperability of databases, etc...) to explain the “Where relevant” introduction of the paragraph.	Thank you for the suggestion. The following text has been added (pg 13): “The comparisons are re-assuring across both the urban and rural sites, validating the indicator outputs of the PER and demonstrating utility to the services. The data will contribute to detailed aggregate reports for facility managers and streamline the monthly submissions to the DHIS which are currently based on manual counts.”
1.7	Page 15 – 3rd paragraph: Replace “In line with WHO recommendations” with “ In line with the TDR protocol”	The change has been applied.
1.8	Page 16 – 1st paragraph under weaknesses: in addition to misclassification, consider under-reporting with the difficulty to establish baseline rates	The following changes have been applied, “The PER digitizes clinical data recorded in operational stationery and we cannot control for data quality nor account for missing data in the source documents, including the risk of under-reporting.”
1.9	Ref [8] for TB drug: the link is not the right one. it is for TPT in pregnant women with HIV then the latest guidelines are the ones from 2020; WHO position has not changed. Should use: https://tbksp.org/en/node/43	The link has been corrected: 8. World Health Organization. Module 1: Prevention, Tuberculosis Preventive Treatment in Consolidated Guidelines on Tuberculosis. Geneva: World Health Organization; 2020 [Available from: https://tbksp.org/en/node/43]